# Bias free multiobjective active learning for materials design and discovery

Kevin Maik Jablonka [1], Giriprasad Melpatti Jothiappan[2], Shefang Wang[2], Berend Smit [1]✉ & Brian Yoo [2]✉

The design rules for materials are clear for applications with a single objective. For most applications, however, there are often multiple, sometimes competing objectives where there is no single best material and the design rules change to finding the set of Pareto optimal materials. In this work, we leverage an active learning algorithm that directly uses the Pareto dominance relation to compute the set of Pareto optimal materials with desirable accuracy. We apply our algorithm to de novo polymer design with a prohibitively large search space. Using molecular simulations, we compute key descriptors for dispersant applications and drastically reduce the number of materials that need to be evaluated to reconstruct the Pareto front with a desired confidence. This work showcases how simulation and machine learning techniques can be coupled to discover materials within a design space that would be intractable using conventional screening approaches.

[1] Laboratory of Molecular Simulation (LSMO), Institut des Sciences et Ingénierie Chimiques, Ecole Polytechnique Fédérale de Lausanne (EPFL), Rue de l'Industrie 17, Sion CH-1951, Switzerland. [2] BASF Corporation, Tarrytown, New York 10591, USA. ✉email: berend.smit@epfl.ch; brian.yoo@basf.com

The holy grail of material science is to find the optimal material for a given application. Finding the optimal material requires a metric to rank the materials. In case we have a single objective, our aim is clearly defined: we evaluate the performance indicator of the materials with respect to this objective and we can rank our materials. Developing efficient strategies to find such an optimum with a minimal number of experiments is an active area of research. In many practical applications, scientists and engineers are often faced with the challenge of having to simultaneously optimize multiple objectives. Optimizing one objective alone may come at the cost of penalizing others[1]. For example, in drug discovery, scientists have to balance potency or activity with toxicities and solubility; or in the field of chemical process design, engineers have to optimize yields for several process units yet sacrifice, say, the energy consumption. Likewise, in the field of material science, desirable material properties can be interdependent or even inversely related. For example, one would like a material that is both strong and ductile, and as these are inversely correlated, it is challenging to synthesize new materials that satisfy both criteria at the same time[2]. In these cases, there is no unique way to rank the materials.

If one has multiple objectives, a practical solution is to combine the different performance indicators into a new overall performance indicator. However, unless such an overall performance indicator is a unique, well-defined function of different performance indicators (e.g., costs), the arbitrary combination of performance parameters obscures the true nature of the optimization problem; there simply is no material that simultaneously optimizes all target properties. No single optimum is generally preferred over all the others; hence, the most valuable information any search in the design space can give is the set of all possible materials for which none of the performance indicators can be improved without degrading some of the other indicators. In statistical terms, these materials are referred to as the set of all Pareto-optimal solutions (i.e., the Pareto front). In this study, we address the question of how to efficiently search for this set of materials and with confidence to not discard a good material. Such a methodology is particularly important if, because of limited resources, it is difficult to evaluate an unlimited number of materials.

Recently, there has been quite some research effort to use machine learning for the design and discovery of new materials[3–7]. A naive approach would be to train a machine learning (surrogate) model to make predictions for all materials in the design space and then use these predictions to compute the Pareto front. However, from a practical point of view, such an approach is not so efficient, as it is not clear how to choose a training set that makes the model confident in the relevant regions of our design space. A random, or even diverse set, will probably contain more points than we actually need and does not consider that we do not need the same accuracy in all parts of our design space. The question we therefore need to answer is how we can efficiently train this model to make confident predictions in the relevant regions of our design space. An appealing way to do this is active learning[8]. Here we initialize a model with a small sample of our design space and then iteratively add labels, i.e., measurements or simulation results, to the training set where the model needs them most. This allows us to efficiently build a model that is able to solve the question of what materials are Pareto optimal and which ones we should discard for further investigation.

It is instructive to compare this approach with Bayesian optimization[9–14]. In such an optimization, one would like to know the next best measurement by typically (and implicitly) assuming that the current evaluation will be the final evaluation[15]. Then, we can use an acquisition function to propose the next best measurement based on the predictions of a machine learning model. This best measurement can then be added to the training set and in this way one can selectively improve the predictions of

the model in a potentially promising part of design space. However, most, if not all, of these optimization techniques rely on the introduction of a total order in the search space with which the materials are ranked in terms of performance. This biases the search (or introduces other technical difficulties, which we discuss in Supplementary Note 1). In this context, it is important to realize that, mathematically speaking, Pareto dominance only defines a partial order in our design space. This means we can only say if a material is Pareto dominating or not, but we cannot directly compare them; hence, the introduction of a total order is nothing more than a (subjective) formula on how to compare apples and pears[16].

In this study, we show how to recover without such bias, but with confidence, a prediction of the Pareto front in the context of polymers discovery. The rational design and discovery of polymers has been a longstanding challenge in the scientific community due to its combinatorial chemical and morphological complexity, which also requires the consideration of multiple spatiotemporal scales[17–19]. In our approach, we use machine learning to predict the next best experiments to systematically reduce the uncertainty of our prediction of the Pareto front until all polymers within our design space can be confidently classified. To reach this goal, we use a modified implementation of the $\epsilon$-PAL algorithm introduced by Zuluaga et al.[20,21], which iteratively reduces the effective design space by discarding those polymers from which we know, with confidence from our model predictions (or measurements), that they are Pareto-dominated by another polymer. To make progress in this search, we evaluate the polymer with the highest dimensionless uncertainty from the set of possible polymer candidates, which our model predicts to be near or at the Pareto optimality. The search terminates when all points are classified as Pareto efficient or discarded. Overall, this method has some additional advantages that can be important for materials design and discovery applications. For example, we show how we can tune the granularity of the approximation to the Pareto front in every objective and, in this way, trade off efficiency with accuracy. Moreover, conventional active learning methods often require complete data sets, whereas in most practical applications we are often faced with a situation where we have a lot of data for one property and much less for another. Our method can deal by construction, with partially missing data in the objective functions, i.e., missing one property measurements for some materials, and also can take into account noise in the measurements. Therefore, given its broad applicability, we anticipate that the same workflow will accelerate the design process in the lab.

## Results

The polymers in our study are representative of dispersants that are typically used in solid suspension systems to prevent the flocculation of suspended particles, e.g., as means to ensure the color strength of pigments in coatings applications[22]. Finding the optimal polymer for a dispersant-based application is a typical example of a multiobjective search. One would like to obtain a polymer that has optimal adhesion strength to the surface of the particles that need to be suspended. Once on the surface, the polymers need to repel the other particles and finally, one needs to ensure that the viscosity of the solution ensures kinetic stability[23]. Interestingly, some of these criteria are in competition with each other. For instance, we can imagine that certain monomer types will enhance both the binding to the surface and the attraction between the polymers. In this case, there is no unique solution and we have to trade binding with the surface with the repulsion between the polymers. This is a general observation in many multiobjective problems. We will often find

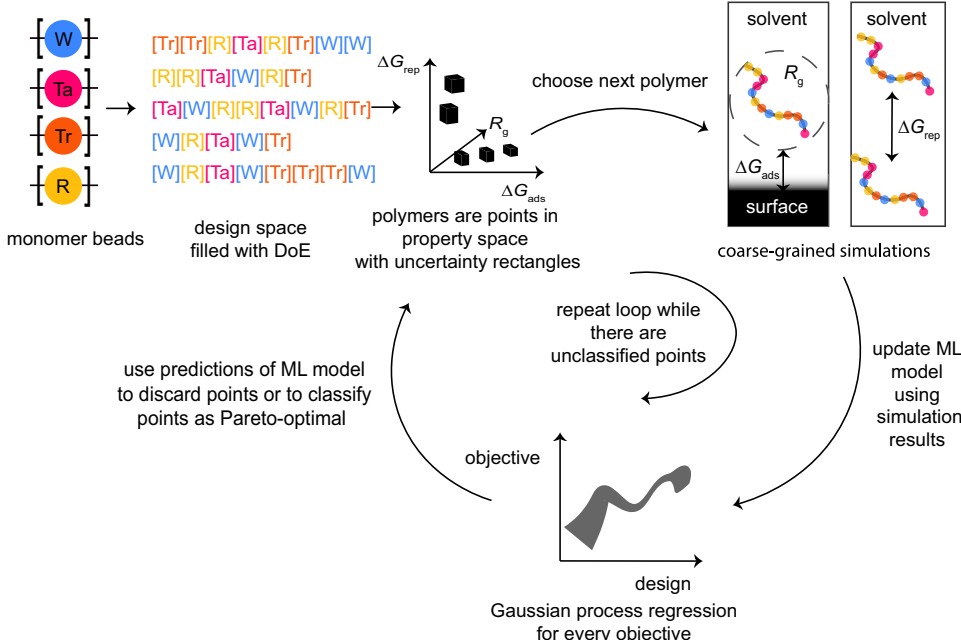

**Fig. 1 Overview of the workflow.** Using classical design of experiments (DoE), we enumerate representative samples in the design space of monomer sequences, which we then explore in the active learning loop with the $\epsilon$-PAL algorithm. For this algorithm, Gaussian process surrogate models provide us with predicted means and standard deviations (SDs) that enable us to decide which designs we can confidently discard, classify as Pareto optimal, and determine which simulation we should run next to maximally reduce the uncertainty for points near the Pareto front. Models that are trained over the course of this process can reveal structure–property relationships and can be inverted using genetic algorithms to further explore the design space.

natural, completely unavoidable competing goals such as the strength-ductility trade-off.

In this work, we mimic the design of our dispersant using a coarse-grained model (see Fig. 1). Our model represents a typical linear copolymer that is often used as a dispersant. In this coarse-grained model, we map monomers with different interactions with the surface and the solvent to different beads, which translates, in our case, to a design space containing more than 53 million possible sequences of polymer beads (see Supplementary Note 2). For a given hypothetical dispersant, we use molecular simulation techniques to evaluate our three ("experimental") key performance indicators. Although we carry out the synthesis and experiments in silico, the number of possible dispersants and the required computational time to evaluate the performance is too large for a brute-force screening of all 53 million dispersants of our coarse-grained polymer genome[24]. Therefore, also for this in silico example, we are limited by our resources and we aim to obtain our set of Pareto-optimal materials as efficiently as possible.

**Dispersants design**. The model polymers investigated in this work are representative of dispersants used in solid suspension systems. That is, each bead in our coarse-grained simulation represents a monomer in a copolymer (Fig. 1). In practice, dispersant performance can be evaluated based on several fundamental driving forces. First, the adhesion strength of the polymer onto a suspended particle surface; second, the steric stability of the polymer, i.e., the ability to help repel suspended particles from one another; finally, the viscosity of the polymer solution which is associated with the kinetic stability of the system[23,25]. To characterize such driving forces, we calculate the following properties (Fig. 2) using coarse-grained dissipative particle dynamics simulations[26]: (i) the adsorption free energy ($\Delta G_{ads}$) onto a model surface, quantifying the adhesive strength to the surface; (ii) the dimer free energy barrier ($\Delta G_{rep}$) between two of

the same polymers, as a metric for the repulsion between the polymers; and finally (iii) the radius of gyration ($R_g$), a molecular property commonly associated with the polymer viscosity[27,28], and which can be experimentally determined using small-angle X-ray scattering[29].

The main objective of this study is to identify polymer sequences that optimize all three of these molecular properties from a sequence design space comprising 4 possible monomer types, with the number of monomers for each type ranging from 4 to 12.

We initially sample our polymer design space (Fig. 3), i.e., the possible arrangements of monomers, by performing full factorial experimental design on the monomer types, where each monomer type contains a selection of monomer counts. This ensures that we enumerate through all possible combinations of available monomer counts and types (see "Methods"). Compared to sampling from the latent space of generative models such as variational autoencoders, this approach maintains a high level of model interpretability and does not require a prior database of structures, which are often used to train autoencoders. Monomer sequences are generated in random order based on these design points. We then explore the space sampled with design of experiments using our active learning algorithm to find the Pareto-optimal polymers. An overview of our workflow is illustrated in Fig. 1.

**Pareto active learning**. In this work, we are interested in not only efficiently, but confidently, identifying an approximation of the Pareto front. To achieve this, we need two ingredients: first, a way to discard points or to classify them as Pareto optimal, and second, a way to propose the next best sample(s) to evaluate. Our modified version of the $\epsilon$-PAL algorithm[20,21] addresses these matters by using the uncertainty estimate ($\sigma$) of a Gaussian process regression surrogate model to construct hyperrectangles for a predicted material (Fig. 4).

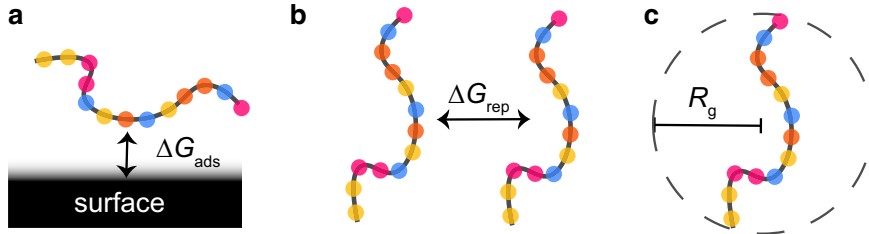

**Fig. 2 Schematic illustration of the polymer performance descriptors that we calculate using coarse-grained simulations. a** $\Delta G_{ads}$ is the single-molecule free energy of adsorption onto a model surface. **b** $\Delta G_{rep}$ is the dimer repulsion energy. **c** $R_g$ is the radius of gyration, an indicator of polymer viscosity.

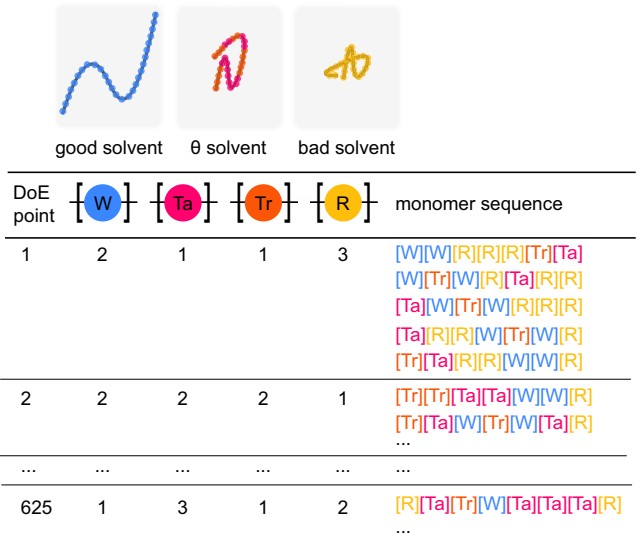

| | good solvent | θ solvent | bad solvent | |
|---|---|---|---|---|

| DoE point | W | Ta | Tr | R | monomer sequence |
|---|---|---|---|---|---|
| 1 | 2 | 1 | 1 | 3 | [W][W][R][R][R][Tr][Ta] |
| | | | | | [W][Tr][W][R][Ta][R][R] |
| | | | | | [Ta][W][Tr][W][R][R][R] |
| | | | | | [Ta][R][R][W][Tr][W][R] |
| | | | | | [Tr][Ta][R][R][W][W][R] |
| 2 | 2 | 2 | 2 | 1 | [Tr][Tr][Ta][Ta][W][W][R] |
| | | | | | [Tr][Ta][W][Tr][W][Ta][R] |
| | | | | | ... |
| ... | ... | ... | ... | ... | ... |
| 625 | 1 | 3 | 1 | 2 | [R][Ta][Tr][W][Ta][Ta][Ta][R] |
| | | | | | ... |

**Fig. 3 Illustration of the DoE approach.** The beads of our coarse-grained model have different interactions with the solvent. The "[W]" bead corresponds to a polymer in a good solvent, the "[R]" bead to a polymer in a bad solvent, and the "[Ta]" and "[Tr]" beads to polymers in a theta solvent. "[Tr]" and "[Ta]" differ from each other in their interaction with the surface. For each DoE point, which specifies the composition of a polymer, we sample five arrangements of monomers. This results in a design space of 3125 polymers in total. Note that the polymers that we sampled had at minimum 4 units of each monomer.

Let us assume we have two objective functions. In Fig. 4, we illustrate the working principle of the $\epsilon$-PAL algorithm. We start with a set of diverse experiments for which we measured the objectives. Based on these experiments, we can train an initial model, using features that are simple to compute and are intuitively related to the chemistry of the polymers (solely based on the monomer sequence), and can make predictions for all the polymers that are indicated as black points. For each point, we construct hyperrectangles, shown in Fig. 4a, around the mean $\mu$ (which comes either from the model predictions or the measurement) with a width that is proportional to the uncertainty $\sigma$ (the SD of the posterior) for the points we did not sample so far and the estimated uncertainty of the measurement for the sampled points (the exact width of the uncertainty hyperrectangles is also a function of the hyperparameters and the iteration, see Supplementary Information for details). The lower and upper limits of these hyperrectangles are the respective pessimistic and optimistic predicted performance estimates for all the objectives.

From the ($\epsilon$)-Pareto dominance relation, we can identify the points that can be discarded with confidence (gray in Fig. 4b) and those of which are with high-probability Pareto optimal (colored blue) as shown in Fig. 4b. If the pessimistic estimate for our

predicted material is greater than a tolerance (defined using the $\epsilon$ hyperparameter) above the optimistic estimate for all other materials, it will be part of the Pareto front. Our current estimate of the Pareto front is then the (thick) blue line connecting the blue points. In addition, we can make a simple estimate of the accuracy of our current prediction of the Pareto front by connecting the bottom left corners of hyperrectangles associated with our current estimate of the front, which gives us the most pessimistic front (lower blue line). The optimistic front is then obtained by connecting the upper right corners. For the case of multiobjective maximization using this algorithm, we can discard materials with high certainty if the optimistic estimate of the material is within some set tolerance ($\epsilon$) below the pessimistic estimate of any other material. We maintain the orange point, as it cannot be discarded within our set uncertainty (see Fig. 4b). Hence, we have a simple geometric construction that allows us to classify whether a predicted material is Pareto optimal or whether we can discard it with certainty.

After this classification, we can, with certainty, discard all experiments of which the hyperrectangles are completely below the most pessimistic front. This significantly reduces our design space. In terms of Bayesian optimization, this can be thought of as the exploitation step.

Following this classification, the next step is to determine the next material to run experiments on. The next material to characterize should be the one that reduces our uncertainty in classifying points as Pareto optimal. For this, we assume that the uncertainties are normalized by the predicted mean such that the area of our hyperrectangles represents the relative error (i.e., we use the coefficient of variation). We then simply improve the information gain of our model the most if we reduce the uncertainty of the largest rectangle among points presumed near or at the Pareto front. In Fig. 4c, the biggest area corresponds to the orange point and adding an extra point will improve the accuracy of our model in that part of the Pareto front. As a result, we obtain a more accurate estimate (see Fig. 4d). We can continue this procedure by sampling the next largest hyperrectangle(s) until our prediction of the Pareto front has reached the desired accuracy. The model is then retrained using all sampled points, including those that have been discarded.

It is interesting to note that all the points we discard are with high probability not part of the $\epsilon$-Pareto front. Hence, we do not need to sample points from this region of design space even though those points may contain the largest uncertainty regions out of the entire set. Interestingly, by choosing the hyperparameters properly, we can also obtain theoretical guarantees on the quality of the Pareto front. That is, given a kernel of a predictive Gaussian process regression (GPR) model and proper scaling parameters of the hyperrectangles, $\epsilon$ will be the maximum error of our Pareto front with probability $\delta$ (see Supplementary Note 10)[20]. Setting a larger tolerance $\epsilon$ will speed up the classification of the design space but increase the errors. In practice, it is reasonable to set $\epsilon$ to be larger than the error of the experiment/simulation.

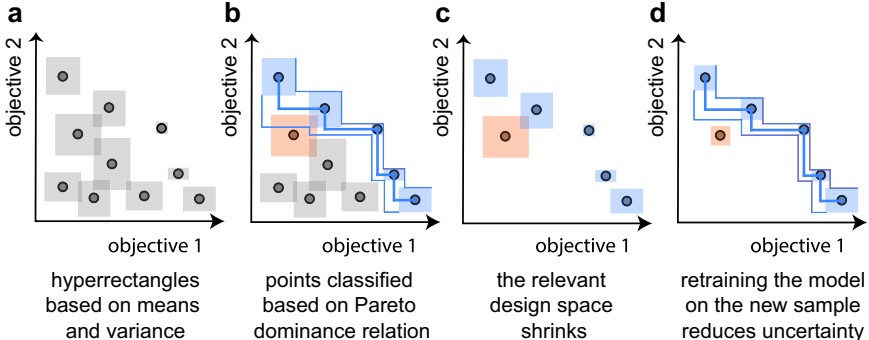

**Fig. 4 Illustration of the working principle of the $\epsilon$-PAL algorithm. a** For each point, we construct hyperrectangles around the mean $\mu$ (coming from either from the model predictions or the measurement) with widths proportional to the uncertainty $\sigma$ (which is the SD of the posterior of the points we did not sample yet and the estimated uncertainty of the measurement for the sampled points; the exact width of the uncertainty hyperrectangles is also a function of the hyperparameters and the iteration). **b** Using the $\epsilon$-Pareto dominance relation, we can identify which points can be discarded with confidence and which are with high-probability Pareto optimal. **c** After this classification, the design space that is relevant for search is smaller and we can sample the largest hyperrectangle to reduce uncertainty (the orange one in this case). **d** After performing the simulations for the sampled material (orange), the model uncertainties decrease, notably in the neighborhood region of the sampled material.

Here we use this algorithm to efficiently choose which simulations to run, although, in principle, one can apply the same algorithm to efficiently choose the experiments—e.g., in self-driving laboratories[30] or in other related multiobjective materials discovery problems, where we would like to recover the Pareto front within some level of granularity $\epsilon$.

For this study, we have performed brute-force simulations and obtained property estimates for all of the design points generated from our DoE approach so that we can evaluate the effectiveness of the algorithm. This allows us to recover the true Pareto front (in the space sampled with DoE) and compare it to our predicted Pareto front obtained after each active learning cycle. Figure 5 presents the property estimates, Pareto-optimal points, and the sampled points in property space.

A key metric for evaluating the quality of the Pareto front is the so-called hypervolume indicator. This indicator measures the size of the space enclosed by the Pareto front and a user-defined reference point (in two dimensions, this would equate to the enclosed area), and is commonly used to benchmark Bayesian optimization algorithms. In general, a better design will always have a larger hypervolume[16]. Using this indicator, we analyze how accurately and rapidly our active learning approach recovers the true Pareto front. In addition, we compare our approach with random sampling. It is noteworthy that random sampling might seem like a naive approach; however, it has been shown to be an efficient search method, e.g., for outperforming grid search in many optimization problems[31]. Hence, it is a relevant baseline.

Figure 6a illustrates the working principle and effectiveness of the algorithm. It attempts to classify the polymers in the design space as fast as possible into either an $\epsilon$-accurate Pareto optimal or to a discarded polymer. Each iteration corresponds to the (in silico) synthesis of a new dispersant and subsequent evaluation of the three key performance indicators: the adsorption free energy ($\Delta G_{ads}$), the dimer free energy barrier ($\Delta G_{rep}$), and the radius of gyration ($R_g$). The data show that already after ten iterations the algorithm confidently discards many polymers (orange region) and finds many $\epsilon$-accurate Pareto-optimal polymers (blue region). In Fig. 6b, we compare the performance of the algorithm with random search and use the hypervolume error—the relative error with respect to the maximum hypervolume of the design space—to quantify the quality of the estimated Pareto front. We can observe that $\epsilon$-PAL achieves the target error ($\epsilon = 0.01$) with >89% fewer iterations compared to random exploration of the design space (153 with our approach, 1421 with random search).

An extension of our approach is a case in which we have missing data. Often in experimental data sets, data are missing for a property that is more difficult to measure. In our case, the calculation of the dimer repulsion energy requires significantly more computational time than the other properties. Hence, it would be interesting to see how such an algorithm performs if, say, 30% of the data is lacking for one of the properties (i.e., one of the properties is immeasurable for some of these materials). Figure 6b presents the performance of the algorithm for when a third of the dimer repulsion energies are missing. In this situation, using independent Gaussian processes for each objective and running a subsequent experiment with a missing datum would not improve model predictions for that property. The idea is that we capture correlations between our various objectives by means of coregionalized Gaussian process models[32]. These models allow us to predict multiple objectives using a single surrogate Gaussian process model and provide us better estimates for missing objectives, if one (or more) of the objectives is missing while all other are present for a given design point (see "Methods").

**Chemical insights**. Interestingly, we can not only use our surrogate models as part of the design loop to expedite the discovery process but we can also obtain some understanding of structure–property relationships.

We use the SHapley Additive exPlanations (SHAP) technique to obtain chemical insights into what the models learned during the discovery process. This method can reveal how the features used by the model influence the predictions and how those features interact with one another[33]. In our case, we use SHAP to understand the structure–property relationships. In Fig. 7, we list the five features that, according to our machine learning model, are most important for every target in order of relevance.

Let us first focus on the radius of gyration (Fig. 7a). The most important feature for the prediction of the radius of gyration is the degree of polymerization, followed by the number of good solvent segments ([W]), the number of bad solvent segments ([R]), the number of theta solvent beads ([Tr]), and the relative entropy of the monomer sequence. From Flory's scaling relation, we know that the radius of gyration scales with the chain length $N$: $R_g \sim N^\nu$, where $\nu$ is the Flory exponent[34]. We find that our model detects this direct proportionality between chain length and the radius of gyration. This showcases that our model captures theoretically consistent relationships during the active learning process. More interestingly, we can see that the SHAP

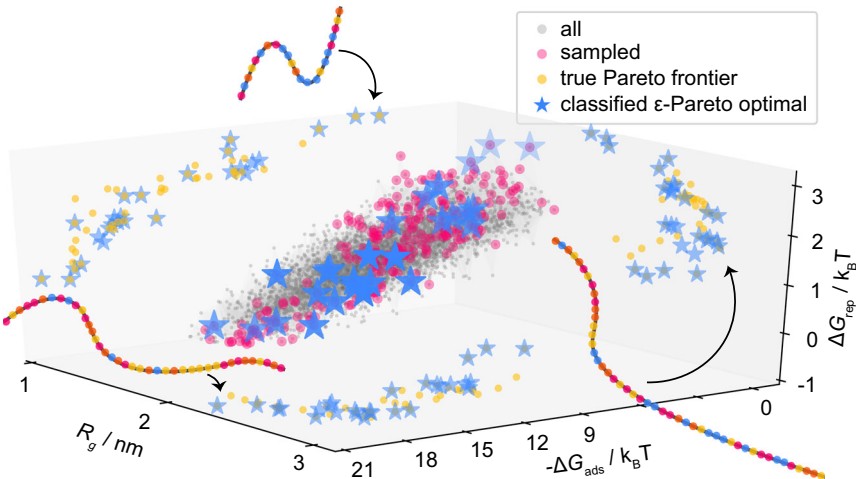

**Fig. 5 Representation of polymers in property space.** Simulations have been performed on the entire experimental design, to determine the three key performance indicators: the adsorption free energy ($\Delta G_{ads}$), the dimer free energy barrier ($\Delta G_{rep}$), and the radius of gyration ($R_g$). Each gray point corresponds to the performance of a unique polymer. Points that have been sampled or classified as $\epsilon$-Pareto optimal by the $\epsilon$-PAL algorithm are marked in magenta and blue, respectively. Pareto-optimal points have also been projected on their respective 2D planes. The schematic drawings of the polymers indicate that the Pareto-optimal materials in our design space have vastly different compositions, e.g., showing a large difference in the degree of polymerization (see Supplementary Note 8).

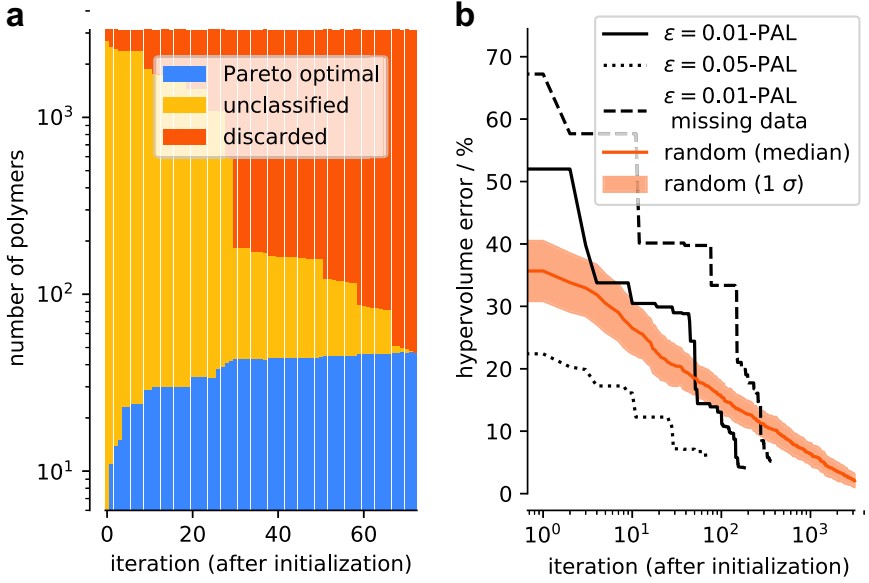

**Fig. 6 Classified points and hypervolume error as a function of the number of iterations. a** The $\epsilon$-PAL algorithm classifies polymers after each learning iteration with $\epsilon_i = 0.05$ for every target and a coregionalized Gaussian process surrogate model. The Gaussian process model was initialized with 60 samples that were selected using a greedy farthest point algorithm within feature space. It is noteworthy that the y-axis is on a log scale. **b** Hypervolume errors are determined as a function of iteration using the $\epsilon$-PAL algorithm with $\epsilon_i = 0.01$ and 0.1 for every target. A larger $\epsilon_i$ makes the algorithm much more efficient but slightly degrades the final performance. For $\epsilon_i = 0.01$, we intentionally leave out a third of the simulation results for $\Delta G_{rep}$ from the entire dataset. The method for obtaining improved predictions for missing measurements with coregionalized Gaussian process models is discussed in more detail in Supplementary Note 7. Hypervolume error for random search with mean and SD error bands (bootstrapped with 100 random runs) is shown for comparison. For the $\epsilon$-PAL algorithm, we only consider the points that have been classified as $\epsilon$-accurate Pareto optimal in the calculation of the hypervolume (i.e., with small $\epsilon$, the number of points in this set will be small in the first iterations, which can lead to larger hypervolume errors). All search procedures were initialized using the same set of initial points, but vary substantially after only one iteration step due to the different hyperparameter values for $\epsilon$. It is noteworthy that the x-axis is on a log scale. Overall, the missing data increase the number of iterations that are needed to classify all materials in the design space.

analysis on the last two features already highlights a key difference between the theory and our model: our model provides us with insights into what happens when we change the composition (e.g., increase the ratio of [W] or [R]). For example, if we increase the fraction of good solvent beads ([W]), we have a higher radius of gyration, whereas the radius decreases if we

increase the number of bad solvent beads ([R]). Hence, our model closely recovers our intuition and additionally, allows us to quantitatively capture these effects.

We use the same machine learning model to predict the two other key performance parameters—the interaction with the surface and repulsion between the dispersants—and also use SHAP to

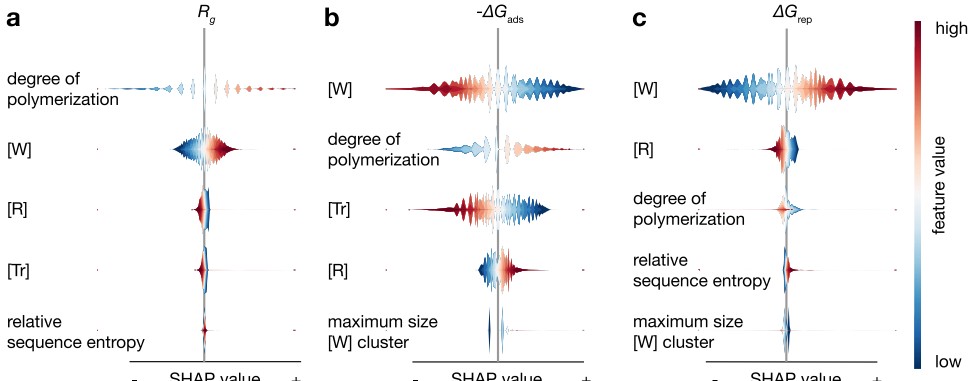

**Fig. 7 Influence of the feature values on the predictions.** SHAP summary plots for the models for our three objectives based on the models obtained after running the $\epsilon$-PAL algorithm in Fig. 6. **a** SHAP summary plot for the radius of gyration. **b** SHAP summary plot for the adsorption energy. **c** SHAP summary plot for the dimer repulsion energies. We used all the sampled points from the run of the $\epsilon$-PAL algorithm as background data for the SHAP analysis. Red points correspond to a high feature value, whereas blue points correspond to low feature values. The width of the violin shown on the x-axis corresponds to the density of the distribution of SHAP values and indicate how the features impact the model output. A negative SHAP value means that the specific feature value decreases the predicted value, with respect to the baseline prediction. SHAP values were computed for a coregionalized model with a Matérn-5/2 kernel and $\epsilon = 0.05$.

extract the feature importance. Here we see that by increasing the ratio of [W], we decrease the interaction with the surface but increase the repulsion between the polymers. Interestingly, we find that for the dimer repulsion energy, increasing the relative sequence entropy of the monomers increases the repulsion between dimers. This implies that if one plans to maximize the repulsion between the polymers, one should increase the disorder of the arrangement of the monomers, i.e., avoid blocks. Importantly, we also see that the feature relevance varies between targets, highlighting why a multiobjective search—in contrast to independent single objective search—is pertinent when aiming to accelerate the multiobjective materials discovery process.

**Inverse design**. To investigate whether our algorithm missed potentially better performing polymers that we did not consider in our experimental design, we invert the machine learning models that were trained on-the-fly during the active learning cycle. To do so, we use elitist genetic algorithms (GAs) to find novel polymer structures that maximize the output of our models, while biasing the generation of polymers to ones that are different from the monomer sequences that we considered in the DoE (by adding explicit novelty terms into the loss function, see Supplementary Note 11). This exploits our machine learning model's ability to capture relevant regularities from the design space.

Figure 8 shows the property distribution of the best performing polymers we found based on the output of the GA compared to our original results. We find that independent of whether we bias the GA towards exploration or exploitation, we cannot find polymers that Pareto-dominate the points that we found using our combination of the DoE and $\epsilon$-PAL approaches.

## Discussion

In materials design, one typically has to balance different objectives and the proper weighting of these objectives is usually not clear in the early design stages. This insight raises the need for a method that can identify the Pareto-optimal points efficiently, while not discarding interesting materials. Using key thermodynamic descriptors derived from molecular simulations for a large polymer design space, we show that our materials design approach can be used to explore polymer genomes that would be intractable using conventional screening methods. Our approach finds the relevant polymers in a fraction of the evaluations that

are needed using traditional approaches and provides us with predictive models and structure–property relationships on-the-fly, while being robust to missing data. This showcases how the coupling between data-driven and conventional materials design approaches, such as simulations or experiments, can greatly enhance the rate with which we discover or optimize materials, while concurrently giving us insights into structure–property relationships.

The vision behind our approach is that in a multiobjective optimization task, the only rigorous result one can obtain is the set of Pareto-optimal materials. Hence, one should focus on an algorithm that systematically improves the accuracy of the estimated Pareto front. Ranking the materials in a multiobjective optimization task introduces, by definition, a bias and detailed studies have been made to identify how such bias can impact the optimization (see ref. [35] and Supplementary Note 1). However, one can make a bias-free ranking of the experiments that improve the accuracy of the Pareto front the most. This observation can be translated into an $\epsilon$-PAL machine learning algorithm and our case study shows that significant gains in efficiency can be achieved.

As multiobjective optimization is such a general problem, we expect that this approach can be adapted to those cases in which efficiency is essential.

## Methods

**Coarse-grained model**. In our model, the polymer bead diameters are assumed to be greater than the Kuhn length, i.e., polymers follow the ideal chain behavior. In total, there are four different polymer bead types in addition to one solvent and two surface bead types. Each bead type, [W]—"weakly attractive," [R]—"repulsive," [Ta]—"theta attractive", and [Tr]—"theta repulsive", was created based on their solvent [S] interaction. Bead types [Ta] and [Tr] are representative of beads for homopolymers in a theta solvent, but differ based on their attractive or repulsive interaction with the surface monolayer bead type [S2]. Bead type [R] is the most adsorptive onto our model surface, whereas bead type [W] is the least attractive. More details on the interaction parameters are provided in Supplementary Note 4.

**Design of experiments**. The first step in the workflow involves the generation of polymers based on our experimental design space. To effectively sample from this design space, we used a full factorial experimental design with the number of factors equal to the number of bead types (4), the number of levels equal to the number of possible bead count variations (5 possible: 4, 6, 8, 10, 12), and 5 unique monomer sequences for each point. Although this is certainly not representative of the entire sequence design space of our polymers, we assume for the purposes of this work that sequence effects come secondary to monomer content. We find this assumption to be a reasonable approximation as noted in Supplementary Note 3.

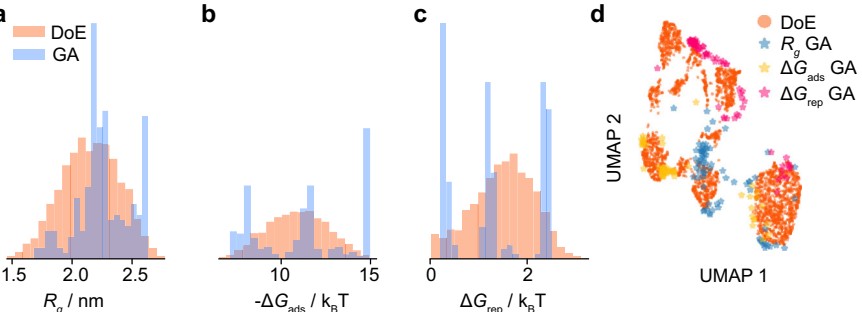

**Fig. 8 Distribution of properties for the polymers found by inverting the machine learning models using a genetic algorithm (GA).** To generate new feature vectors, we ran the genetic algorithm with different weights for the novelty part of the loss function—ranging from no penalty for similar polymers to high (50 times the objective) penalty for polymers similar to the ones already sampled with our design of experiments (DoE). In addition, we run the genetic algorithm for different elitist ratios. For each feature vector, three possible polymer bead sequences were generated, as a feature vector does not map to one unique arrangement of monomers. **a** Distribution of radius of gyration ($R_g$). **b** Distribution of adsorption energies $\Delta G_{ads}$. **c** Distribution of dimer repulsion energies $\Delta G_{rep}$. **d** Polymer properties obtained from GA are projected onto the first two uniform manifold approximation and projection (UMAP) components and are compared to those obtained from DoE. See Supplementary Fig. 30 for principal component analysis.

Overall, we obtain a total of 3125 unique linear polymer molecules represented by their monomer sequence. Experimental design was created using pyDOE[36].

**Simulation protocol**. All simulations were set up using Enhanced Monte Carlo (EMC) version 9.4.4[37,38] and were run with Large-scale Atomic/Molecular Massively Parallel Simulator (LAMMPS) version 2018/03/16[39]. Monomer sequences for coarse-grained polymers were directly ported into LAMMPS input files using EMC.

*Free energy calculations*. Free energy calculations were performed using the LAMMPS plugin provided in Software Suite for Advanced General Ensemble Simulations version 0.82[40]. Steered molecular dynamics simulations[41] were first performed to generate initial configurations for each polymer–surface or polymer–polymer center of mass separation distances of a given polymer[26]. Umbrella sampling was subsequently performed and analyzed using the weighted histogram analysis method[42], to estimate both the adsorption-free energies of the dispersants onto the model dispersion surface and the dimer-free energies of dispersants.

**Machine learning**
*Featurization*. We calculated features such as the degree of polymerization, the relative sequence entropy, the nature of the end groups (one-hot encoded), summed interaction parameters, and the nature of clusters based on the monomer sequence.

All features were $z$-score standardized using the mean and SD of the training set (using the scikit-learn Python package[43]). More details can be found in Supplementary Note 6.

*Gaussian process regression surrogate models*. Intrinsic coregionalized Gaussian process regression models (ICM)[32] (of rank 1) were built using the GPy Python library[44] based on Matérn-5/2 kernels. In Supplementary Note 7, we show that coregionalization improves the predictive performance in the low-data regime, i.e., the initial setting of the algorithm. The ICM models assume that the outputs are scaled samples from the same GPR (rank 1) or weighted sum of $n$ latent functions (rank $n$). A higher rank is connected to more hyperparameters and typically makes the model more difficult to optimize. We provide a performance comparison of rank 1 and rank 2 models in Supplementary Note 7. Hyperparameter optimization was performed with random restarts and in regular intervals as training points were added. More details can be found in Supplementary Note 7. The predictive performance of the models is illustrated in Supplementary Fig. 12.

*Feature importance analysis*. We used the SHAP technique marginalized over the full DoE dataset (summarized with weighted $k = 40$ means-clustering) to calculate model interpretations[33] and the full DoE dataset to calculate SHAP values. For the GPR surrogate models, we apply the "KernelExplainer" method. Model interpretations for runs with different $\epsilon$ are qualitatively consistent, the plot shown in the main text was computed for $\epsilon = 0.05$, and a coregionalized model with Matérn-5/2 kernel.

*Pareto active learning*. We implemented a modified version $\epsilon$-PAL algorithm[20] in our Python package, PyePAL. Our algorithm differs from the original $\epsilon$-PAL algorithm by using the coefficient of variation as the uncertainty measure rather than the predicted SDs. Moreover, our implementation does not assume that the ranges ($r_i$) of the objectives are known. That is, instead using $\epsilon_i \cdot r_i$ for the

computation of the hyperrectangles, we use $\epsilon_i \cdot |\mu_i|$ (see Supplementary Note 10). PyePAL generalizes to an arbitrary number of dimensions as opposed to the original MATLAB code provided by Zuluaga et al.[45] (limited to 2) and, by default, sets the uncertainty of labeled points to the experimental uncertainty or the modeled uncertainty. In addition to supporting standard and coregionalized Gaussian processes surrogate models, our library interfaces with other popular modeling techniques with uncertainty quantification such as quantile regression and neural-tangent kernels. It also offers native support for missing data, e.g., when using coregionalized Gaussian processes, support for both single point (as done in this work) and batch sampling, and the option to exclude high variance points from the classification stage.

Implementation details and hyperparameter settings in this work are provided in Supplementary Note 10. Initial design points used to train the zeroth iteration model were selected using greedy farthest point sampling in feature space[46]. Hypervolume errors shown in the main text were calculated using the nadir point as our reference point.

Our code makes use of the following Python packages: GPy[44], jupyter[47], lightgbm[48], matplotlib[49], neural-tangent[50], nevergrad[51], numba[52], numpy[53], pandas[54], scipy[55], and scikit-learn[43].

*Inverting the GPR models*. To invert the GPR model, we trained Gradient Boosted Decision Tree surrogate models with reduced feature set (e.g., dropping the relative entropy of the monomer sequence) on the predictions of the GPR models. An elitist GA[56] was then used to maximize the output of the model, while being penalized for creating invalid polymer features, i.e., features that cannot be converted to a valid monomer sequence using a backtracing algorithm or features that are very similar to those already present in our dataset. More details can be found in Supplementary Note 11.

## Data availability
The input files for the molecular simulations and the analysis results of the simulations are available on the Materials Cloud[57] Archive (https://doi.org/10.24435/materialscloud:8m-6d). Correspondence and requests for additional materials should be addressed to brian.yoo@basf.com or berend.smit@epfl.ch.

## Code availability
Code for the machine learning part (including the featurization and genetic algorithm) of this study is available as part of the dispersant_screener Python package (archived on Zenodo 10.5281/zenodo.4256868 and developed on GitHub github.com/byooooo/dispersant_screening_PAL). A general-purpose implementation of the $\epsilon$-PAL algorithm, which can be used with other models such as quantile regression, is available as the PyePAL package (archived on Zenodo 10.5281/zenodo.4209470 and developed on GitHub github.com/kjappelbaum/pyepal).

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

## Acknowledgements

K.M.J. and B.S. were supported by the European Research Council (ERC) under the European Union's Horizon 2020 research and innovation program (grant agreement 666983, MaGic), by the NCCR-MARVEL, funded by the Swiss National Science Foundation, and by the Swiss National Science Foundation under grant 200021_172759. The work of G.M.J. and part of the work of K.M.J. were done during the Explore Together internship program at BASF. All molecular simulations were performed on the BASF supercomputer "Quriosity".

## Author contributions

K.M.J., S.W., and B.Y. developed the multiobjective active learning framework. G.M.J. and B.Y. performed the molecular simulations. All authors contributed to the ideation and design of the work. K.M.J., B.S., and B.Y. wrote the manuscript with input from all the co-authors.

## Competing interests

B.Y. and K.M.J. are inventors on a provisional patent application (number 63/108768) describing the use of this algorithm for materials design. The remaining authors declare no competing interests.
