## [Peer Review File · Nature Communications]

REVIEWER COMMENTS

Reviewer #1 (Remarks to the Author):

Identification of optimal materials in multiobjective optimization problems is one of the central challenges encountered in most materials design and discovery efforts. The manuscript presents an active learning strategy to efficiently identify the set of Pareto optimal materials within a targeted accuracy for design problems which focus more than one target attributes/properties (that may have conflicting trends). The method is illustrated using a specific design problem pertaining to dispersant applications. This is a very carefully planned, well executed and clearly written paper with all the relevant technical details provided in the supplemental information document accompanying the manuscript. I have following comments:

Major points:

In general, any active learning based strategy targets to balance the exploration versus exploitation tradeoffs while iterating over the adaptive design loop. At the initial stages, exploration is emphasized over exploitation to ensure a reasonable predictive accuracy of the employed surrogate ML model towards target property prediction for candidate materials coming from different regions of the chemical space. Subsequently, the emphasis is gradually shifted towards exploiting the trained model to push the underlying Pareto front or optimize a prespecified figure-of-merit. It would be very helpful if the authors can consider commenting on how this balancing is carried out in their proposed algorithm.

It is mentioned that an initial set of 60 samples was used to train a Gaussian process regression surrogate model. How exactly these 60 points are selected. Does performance of the final model depend sensitively on the initial number and the specific choice of the elements in this set? In a general problem, what criterion should be used to select this initialization set?

In the proposed algorithm, selection of the next candidate material during the adaptive design loop is carried out by reducing the uncertainty of the largest rectangle among points falling near or at Pareto front. Therefore, this criterion is based on uncertainty reduction on one specific sample that exhibits the lowest prediction confidence (or the largest scaled errorbars). An alternative measure could be based on reducing the overall uncertainty averaged over all the samples that contribute to the predicted current Pareto optimal set at any given iteration during the active learning procedure. Can author comment on the relative merits of the two approaches or why the former approach should be chosen in lieu of the latter?

Minor points:

Redundant sentence:

Page 10: "We maintain the orange point as it cannot be discarded within our set uncertainty, see Fig. 4b."

Page 11: "We maintain the orange one as it cannot be discarded within our set uncertainty, see Fig. 4c."

In Fig. 3, the illustrated configurations under the SMILES column do not match with the number (composition) of beads shown in the table. More specifically, it seems as though the numbers under the R_Ta and R_R columns need to be swapped.

Reviewer #2 (Remarks to the Author):

The authors of "Bias free multiobjective active learning for materials design and discovery" develop a method for efficiently identifying the Pareto front and apply it to a polymer design problem. Notable points include (1) the use of a polymer-based example as the use of ML in polymers is not as rich as in other fields, (2) highlighting the next steps in the design process that are enabled (i.e. inverse design), (3) addressing the issue of explainability in ML through SHAP, and (4) providing the code and data to improve reproducibility and spur other scientific developments. Overall, the manuscript is interesting and

seems to be a good fit for Nature Communications. However, I cannot in good faith recommend the manuscript for publication unless it puts itself in the larger context of the literature. Specifically, epsilon-PAL is not mentioned at all in the introduction, where it should be discussed in the context of prior work. Additionally, the section on Pareto active learning also needs to clarify the exact points that distinguish their method from traditional epsilon-PAL. In addition to this major issue, there are number of minor issues both scientific and clarity related (see below).

Scientific

- More information is needed to support the following sentence as it is not at all obvious how it contributes to model interpretability. "This ensures that we enumerate through all possible combinations of available monomer counts and types, while also maintaining model interpretability (see Methods)." At the minimum, the exact location in Methods is required.
- How are correlations between the uncertainties resulting from three design quantities (R_g , ΔG_{rep} , ΔG_{ads}) handled? Are they assumed to be uncorrelated? How does this fit into the GPR?
- Please provide information on how one should select epsilon. How does it affect performance?
- On a related note, why would a smaller epsilon lead to a bigger error? Please discuss.
- Does the surrogate model keep all the information about data points away from the Pareto front? If not, in principle, the model uncertainty could grow with iteration and you are throwing away useful information.
- For the inverse design part, were any data points found beyond the Pareto front including uncertainty? If so, by how much? Both Fig. 8D and Fig. S22 do not prove things one way or another.

Clarity

- There seems to be an error in the discussion of strong and tough. I believe the authors meant strength and ductility.
- SMILES strings are not actually SMILES strings (line notation that corresponds to actual chemistry as opposed to a coarse-grained system). Please don't use this name as it implies atomistic.
- The beads do not correspond to the chemistry at the left most part of Fig. 1/ top of Fig. 3. Specifically, the beads are not functionalized polypropylene. Please remove.
- Fig. 3 has errors (e.g. there are 3 [R] in DoE point 1 not 1).
- Rephrase "For the case of multiobjective maximization under this algorithm, if the optimistic estimate of our predicted material is within some set tolerance(ϵ) below the pessimistic estimate of any other material (in multidimensional space), we can discard these materials with high certainty." It is confusing as written. It took multiple readings and staring at the figure to determine what was meant.
- Define epsilon sooner, say in paragraph 3, of "Pareto active learning" where tolerance is discussed.
- "This allows us to recover the true Pareto front and compare it to our predicted Pareto front obtained after each active learning cycle." This is an overstatement. Instead, it recovers the Pareto front within the DoE approach, not in general since the method only samples points from the DoE.
- "A key metric for evaluating the quality of the Pareto front is the so called hypervolume indicator, which measures the hypervolume of the objective space, i.e., the size of the space enclosed by the Pareto front and a user-defined reference point (in 2D, this is an area). A better design will always have a larger hypervolume." Please rephrase; it is confusing as written. And specify the reference point you are using; this is important for reproducibility.
- "We can observe that e-PAL achieves the target error (ϵ) with more than 98% fewer iterations compared to random exploration of the design space (11 with our approach, 509 with random search)." You need to say that you use $\epsilon = 0.1$ for your approach.
- "Combined with the DoE, we reduced the number of evaluations from possibly over 53 million (the full polymer design space) to 71 (60 initialization points and 11 iterations until we reached 5% hypervolume error)." This is misleading since you define the hypervolume reference. 5% isn't meaningful. Thus, the sentence should be removed.
- There is an error in Fig. 6. The caption mentions $\epsilon = 0.05$, but the figure doesn't contain the data.
- Which epsilon does Fig. 7 correspond to? The methods say 0.05. If that is correct, it should be in the caption.

RESPONSE TO THE REVIEWERS

In addition to the changes requested by the reviewer we also made some editorial changes in the main text and the supplementary material. All additions to the PyePAL package that we have made for this revision are included in the 0.6.0 release.

REVIEWER 1

Identification of optimal materials in multiobjective optimization problems is one of the central challenges encountered in most materials design and discovery efforts. The manuscript presents an active learning strategy to efficiently identify the set of Pareto optimal materials within a targeted accuracy for design problems which focus more than one target attributes/properties (that may have conflicting trends). The method is illustrated using a specific design problem pertaining to dispersant applications. This is a very carefully planned, well executed and clearly written paper with all the relevant technical details provided in the supplemental information document accompanying the manuscript. I have following comments:

Major points

Reviewer Point P 1.1 — In general, any active learning based strategy targets to balance the exploration versus exploitation tradeoffs while iterating over the adaptive design loop. At the initial stages, exploration is emphasized over exploitation to ensure a reasonable predictive accuracy of the employed surrogate ML model towards target property prediction for candidate materials coming from different regions of the chemical space. Subsequently, the emphasis is gradually shifted towards exploiting the trained model to push the underlying Pareto front or optimize a prespecified figure-of-merit. It would be very helpful if the authors can consider commenting on how this balancing is carried out in their proposed algorithm.

Reply: This question relates to the difference between active learning and Bayesian optimization that we discuss in Supplementary Note 1. Active learning is typically focused on increasing some information criterion, e.g., by reducing the uncertainty, whereas Bayesian optimization relies on optimizing a well-defined (and continuous) acquisition function that balances exploitation and exploration (often with the implicit notion that the next experiment will be the final one). The reason the ϵ -PAL algorithm is practical for materials design/discovery applications is that it focuses on identifying the relevant regions of the design space through classification under uncertainty (without the implicit notion of the next experiment being the final one). The classification step, i.e., discarding the points from which we know with certainty that they are dominated by other points, can be thought of as an exploitation step, and the sampling step (where we sample the most uncertain points from the unclassified and Pareto-optimal set) as exploration step.

To clarify, we now write in the main text

After this classification, we can with certainty discard all experiments of which the hyperrectangles are completely below the most pessimistic front. This significantly reduces our design space. In terms of Bayesian optimization, this can be thought of as the exploitation step.

Reviewer Point P 1.2 — It is mentioned that an initial set of 60 samples was used to train a Gaussian process regression surrogate model. How exactly these 60 points

are selected. Does performance of the final model depend sensitively on the initial number and the specific choice of the elements in this set? In a general problem, what criterion should be used to select this initialization set?

Reply: The initial number of points should be chosen such that the surrogate model is predictive. This can be estimated using cross-validation and learning curve analysis for which we now added a utility in the PyePAL package as well as an example notebook.

Regarding the selection method, we compared greedy farthest point sampling and k-means clustering (for both of which we already provide utilities in our package) given that our previous work has shown the utility of a diverse set selection.¹ We discuss the influence of the number of initial points in detail in Supplementary Note 9.1 and also added a remark to the methods section that reads

Initial design points used to train the zeroth iteration model were selected using greedy farthest point sampling in feature space.²

Reviewer Point P 1.3 — In the proposed algorithm, selection of the next candidate material during the adaptive design loop is carried out by reducing the uncertainty of the largest rectangle among points falling near or at Pareto front. Therefore, this criterion is based on uncertainty reduction on one specific sample that exhibits the lowest prediction confidence (or the largest scaled errorbars). An alternative measure could be based on reducing the overall uncertainty averaged over all the samples that contribute to the predicted current Pareto optimal set at any given iteration during the active learning procedure. Can author comment on the relative merits of the two approaches or why the former approach should be chosen in lieu of the latter?

Reply: In the Supplementary Information we now report the results for some additional experiments with different aggregation functions.

In the PyePAL package we added an option to customize the aggregation function, e.g, to switch from the \mathcal{L}_2 norm to the median or a simple average.

The choice between the different aggregation functions (to combine the uncertainties in different objectives into one scalar) is comparable to the choice one has to make with respect to the loss function in any optimization problem: In some circumstances it can be beneficial to have a high penalty on outliers (mean-squared error) whereas in other circumstances one does not want to have such a penalty (mean-absolute error). We show some case studies in Supplementary Figure 26.

In the Supplementary Note 10 we now also explain why we chose to implement uncertainty sampling

We chose to not implement sampling methods that require retraining of the models for all potential candidates (e.g., expected error reduction^{3,4}) as those techniques would extremely increase the computational cost of the algorithm (retraining and evaluating the model(s) for every possible new sample, averaged over all possible labels), even though those techniques might mitigate the tendency of uncertainty sampling⁵ to sample outliers.

Minor points:

Reviewer Point P 1.4 — Redundant sentence: Page 10: "We maintain the orange point as it cannot be discarded within our set uncertainty, see Fig. 4b." Page 11: "We maintain the orange one as it cannot be discarded within our set uncertainty, see Fig. 4c."

Reply: We deleted the second sentence.

Reviewer Point P 1.5 — In Fig. 3, the illustrated configurations under the SMILES column do not match with the number (composition) of beads shown in the table. More specifically, it seems as though the numbers under the R_Ta and R_R columns need to be swapped.

Reply: We updated the figure in the revised version.

REVIEWER 2

The authors of “Bias free multiobjective active learning for materials design and discovery” develop a method for efficiently identifying the Pareto front and apply it to a polymer design problem. Notable points include (1) the use of a polymer-based example as the use of ML in polymers is not as rich as in other fields, (2) highlighting the next steps in the design process that are enabled (i.e. inverse design), (3) addressing the issue of explainability in ML through SHAP, and (4) providing the code and data to improve reproducibility and spur other scientific developments. Overall, the manuscript is interesting and seems to be a good fit for Nature Communications.

Reviewer Point P 2.1 — However, I cannot in good faith recommend the manuscript for publication unless it puts itself in the larger context of the literature. Specifically, epsilon-PAL is not mentioned at all in the introduction, where it should be discussed in the context of prior work. Additionally, the section on Pareto active learning also needs to clarify the exact points that distinguish their method from traditional epsilon-PAL. In addition to this major issue, there are number of minor issues both scientific and clarity related (see below).

Reply: We now also added references to the work from Zuluaga et al. in the introduction, which now reads

To reach this goal, we use a modified implementation of the ϵ -PAL algorithm introduced by Zuluaga et al.,^{6,7} which iteratively reduces the effective design space by discarding those materials from which we know, with confidence from our model predictions (or measurements), that they are Pareto-dominated by another material.

We discussed algorithmic differences in more detail in the documentation of the package (<https://pyepal.readthedocs.io/en/latest/background.html>) but now added discussion of this to the method section, which now reads

We implemented a modified version ϵ -PAL algorithm⁶ in our Python package, PyePAL. Our algorithm differs from the original ϵ -PAL algorithm by using the coefficient of variation as the uncertainty measure rather than the predicted standard deviations. Moreover, our implementation does not assume that the ranges (r_i) of the objectives are known. This is, instead using $\epsilon_i \cdot r_i$ for the computation of the hyperrectangles, we use $\epsilon_i \cdot |\mu_i|$ (see Supplementary Note 10). PyePAL generalizes to an arbitrary number of dimensions as opposed to original MATLAB code provided by Zuluaga et al.⁸ (limited to 2), and by default sets the uncertainty of labeled points to the experimental uncertainty or the modeled uncertainty. In addition to supporting standard and coregionalized Gaussian processes surrogate models, our library interfaces with other popular modeling techniques with uncertainty quantification such as quantile regression and neural tangent kernels. It also offers native support for missing data, for example, when using coregionalized Gaussian processes, support for both single point (as

done in this work) and batch sampling, and the option to exclude high variance points from the classification stage.

Scientific

Reviewer Point P 2.2 — More information is needed to support the following sentence as it is not at all obvious how it contributes to model interpretability. “This ensures that we enumerate through all possible combinations of available monomer counts and types, while also maintaining model interpretability (see Methods).” At the minimum, the exact location in Methods is required.

Reply: We try to contrast the approach based on the enumeration of beads to generative techniques such as autoencoders. To clarify, we now write

This ensures that we enumerate through all possible combinations of available monomer counts and types (see Methods). Compared to sampling from the latent space of generative models such as standard autoencoders or variational autoencoders, this approach maintains a high level of model interpretability.

Reviewer Point P 2.3 — How are correlations between the uncertainties resulting from three design quantities (Rg, DeltaGrep, DeltaGads) handled? Are they assumed to be uncorrelated? How does this fit into the GPR?

Reply: As indicated in reply to point P 2.1 we implemented a range of different methods for uncertainty estimation such as Gaussian process regression, quantile regression using gradient boosted decision trees or neural tangent kernels in the PyePAL package. Many of those models do not offer native support for multioutput problems and in those cases, the different objectives are modeled independently from each other with separate models. Since objectives usually are correlated with each other it can be useful to take this correlation into account. This is especially important in the case of missing data in the objectives where modeling the correlation between the objectives can improve our ability to “impute” the missing observations. In the case of materials discovery, this can be the case when one experiment/simulation is much more expensive than the ones for other objectives. In the article, we use coregionalized Gaussian process models to address this problem. We explain the assumptions of these models in the revised methods section

The ICM models assume that the outputs are scaled samples from the same GPR (rank 1) or weighted sum of n latent functions (rank n). A higher rank is connected to more hyperparameters and typically makes the model more difficult to optimize. We provide a performance comparison of rank 1 and rank 2 models in Supplementary Note 7.2.

Additionally, we updated the Supplementary Information with experiments using rank=2 ICM models.

Reviewer Point P 2.4 — Please provide information on how one should select epsilon. How does it affect performance?

Reply: There a few considerations when choosing ϵ we now discuss those in the “Pareto active learning” section

Setting a larger tolerance ϵ will speed up the classification of the design space but increase the errors. In practice, it is reasonable to set ϵ to be larger than the error of the experiment/simulation.

Reviewer Point P 2.5 — On a related note, why would a smaller epsilon lead to a bigger error? Please discuss.

Reply: We realize that the caption of Figure 6 is not the best place for this information, but we did not find a better place in the main text. The reason why the initial error can be higher is due to the way we measure the hypervolume error—we only consider the materials we classified as ϵ -accurate Pareto optimal. With a larger ϵ more materials fall initially into this class, whereas it might take longer for the model with smaller ϵ to reach a sufficiently small standard deviation to classify points as Pareto optimal—which then can lead to an initially higher hypervolume error.

We hope to clarify this in the revised version of the caption which reads

Classified points and hypervolume error as a function of the number of iterations. a The ϵ -PAL algorithm classifies polymers after each learning iteration with $\epsilon_i = 0.05$ for every target and a coregionalized Gaussian process surrogate model. The Gaussian process model was initialized with 60 samples that were selected using a greedy farthest point algorithm within feature space. Note that the y-axis is on a log scale. b Hypervolume errors are determined as a function of iteration using the ϵ -PAL algorithm with $\epsilon_i = 0.01$, and 0.1 for every target. A larger ϵ_i makes the algorithm much more efficient but slightly degrades the final performance. For $\epsilon_i = 0.05$, we intentionally leave out a third of the simulation results for ΔG_{rep} from the entire data set. The method for obtaining improved predictions for missing measurements with coregionalized Gaussian process models is discussed in more detail in Supplementary Note 7.3. Hypervolume error for random search with mean and standard deviation error bands (bootstrapped with 100 random runs) is shown for comparison. For the ϵ -PAL algorithm we only consider the points that have been classified as ϵ -accurate Pareto optimal in the calculation of the hypervolume (i.e., with small ϵ the number of points in this set will be small in the first iterations, which can lead to larger hypervolume errors). All search procedures were initialized using the same set of initial points, but vary substantially after only one iteration step due to the different hyperparameter values for ϵ . Note that the x-axis is on a log scale. Overall, missing data increases the number of iterations that are needed to classify all materials in the design space.

Reviewer Point P 2.6 — Does the surrogate model keep all the information about data points away from the Pareto front? If not, in principle, the model uncertainty could grow with iteration and you are throwing away useful information

Reply: The surrogate model keeps all the information about all sampled points. We clarify this now in the “Pareto active learning” section of the main text

The model is then retrained using all sampled points, including those that have been discarded.

Reviewer Point P 2.7 — For the inverse design part, were any data points found beyond the Pareto front including uncertainty? If so, by how much? Both Fig. 8D and Fig. S22 do not prove things one way or another.

Reply: To clarify this point we now write in the inverse design section

We find that independent of whether we bias the GA towards exploration or exploitation, we cannot find polymers that Pareto-dominate the points that we found using our combination of the DoE and ϵ -PAL approaches.

Clarity

Reviewer Point P 2.8 — There seems to be an error in the discussion of strong and tough. I believe the authors meant strength and ductility.

Reply: We thank the reviewer for pointing this out. The sentence in the introduction now reads

For example, one would like a material that is both strong and ductile and as these are correlated it is challenging to synthesize new materials that satisfy both criteria at the same time.⁹

Reviewer Point P 2.9 — SMILES strings are not actually SMILES strings (line notation that corresponds to actual chemistry as opposed to a coarse-grained system). Please don't use this name as it implies atomistic.

Reply: We now replaced all occurrences of "SMILES" with "monomer/bead sequence".

Reviewer Point P 2.10 — The beads do not correspond to the chemistry at the left most part of Fig. 1/ top of Fig. 3. Specifically, the beads are not functionalized polypropylene. Please remove.

Reply: We updated the figure in the revised version of the manuscript.

Reviewer Point P 2.11 — Fig. 3 has errors (e.g. there are 3 [R] in DoE point 1 not 1).

Reply: We updated the figure in the revised version of the manuscript.

Reviewer Point P 2.12 — Rephrase "For the case of multiobjective maximization under this algorithm, if the optimistic estimate of our predicted material is within some set tolerance(ϵ) below the pessimistic estimate of any other material (in multidimensional space), we can discard these materials with high certainty." It is confusing as written. It took multiple readings and staring at the figure to determine what was meant.

Reply: We rephrased to

For the case of multiobjective maximization using this algorithm, we can discard materials with high certainty if the optimistic estimate of the material is within some set tolerance (ϵ) below the pessimistic estimate of any other material.

Reviewer Point P 2.13 — Define epsilon sooner, say in paragraph 3, of "Pareto active learning" where tolerance is discussed.

Reply: We now write in paragraph 3 of "Pareto active learning":

From the (ϵ)-Pareto dominance relation, we can identify those points that can be discarded with confidence (gray in Fig. 4b) and those which are with high probability Pareto optimal (colored blue) as shown in Fig. 4b. If the pessimistic estimate for our predicted material is greater than a tolerance (defined using the ϵ hyperparameter) above the optimistic estimate for all other materials, it will be part of the Pareto front.

Reviewer Point P 2.14 — “This allows us to recover the true Pareto front and compare it to our predicted Pareto front obtained after each active learning cycle.” This is an overstatement. Instead, it recovers the Pareto front within the DoE approach, not in general since the method only samples points from the DoE.

Reply: We now write instead

This allows us to recover the true Pareto front (in the space sampled with DoE) and compare it to our predicted Pareto front obtained after each active learning cycle.

Reviewer Point P 2.15 — “A key metric for evaluating the quality of the Pareto front is the so called hypervolume indicator, which measures the hypervolume of the objective space, i.e., the size of the space enclosed by the Pareto front and a user-defined reference point (in 2D, this is an area). A better design will always have a larger hypervolume.” Please rephrase; it is confusing as written. And specify the reference point you are using; this is important for reproducibility.

Reply: We rephrased this to

A key metric for evaluating the quality of the Pareto front is the so called hypervolume indicator. This indicator measures the size of the space enclosed by the Pareto front and a user-defined reference point (in 2D, this would equate to the enclosed area), and is commonly used to benchmark Bayesian optimization algorithms. In general, a better design will always have a larger hypervolume.

The reference point is now indicated in the methods section.

Reviewer Point P 2.16 — “We can observe that e-PAL achieves the target error (ϵ) with more than 98% fewer iterations compared to random exploration of the design space (11 with our approach, 509 with random search).” You need to say that you use $\epsilon = 0.1$ for your approach.

Reply: We added this information to the sentence.

Reviewer Point P 2.17 — “Combined with the DoE, we reduced the number of evaluations from possibly over 53 million (the full polymer design space) to 71 (60 initialization points and 11 iterations until we reached 5% hypervolume error).” This is misleading since you define the hypervolume reference. 5% isn’t meaningful. Thus, the sentence should be removed.

Reply: We removed the sentence in the revised version of the manuscript and added a discussion of the influence of the hypervolume reference point and target error to the revised SI.

Reviewer Point P 2.18 — There is an error in Fig. 6. The caption mentions $\epsilon = 0.05$, but the figure doesn’t contain the data.

Reply: We fixed the typo.

Reviewer Point P 2.19 — Which epsilon does Fig. 7 correspond to? The methods say 0.05. If that is correct, it should be in the caption.

Reply: We added this information to the caption of the revised version.

REFERENCES

- [1] Moosavi, S. M.; Nandy, A.; Jablonka, K. M.; Ongari, D.; Janet, J. P.; Boyd, P. G.; Lee, Y.; Smit, B.; Kulik, H. J. *Nat. Commun.* **2020**, *11*, 4068.
- [2] Kennard, R. W.; Stone, L. A. *Technometrics* **1969**, *11*, 137–148.
- [3] Roy, N.; McCallum, A. Toward Optimal Active Learning through Sampling Estimation of Error Reduction. Proceedings of the Eighteenth International Conference on Machine Learning. San Francisco, CA, USA, 2001; p 441–448.
- [4] Cohn, D. A.; Ghahramani, Z.; Jordan, M. I. Active Learning with Statistical Models. Proceedings of the 7th International Conference on Neural Information Processing Systems. Cambridge, MA, USA, 1994; p 705–712.
- [5] Lewis, D. D.; Gale, W. A. *SIGIR '94*; Springer London, 1994; pp 3–12.
- [6] Zuluaga, M.; Krause, A.; Püschel, M. *J. Mach. Learn. Res.* **2016**, *17*, 1–32.
- [7] Zuluaga, M.; Sergent, G.; Krause, A.; Püschel, M. Active Learning for Multi-Objective Optimization. Proceedings of the 30th International Conference on Machine Learning. Atlanta, Georgia, USA, 2013; pp 462–470.
- [8] Vivek Nair, epsilon-PAL. <https://github.com/FlashRepo/epsilon-PAL>, 2017.
- [9] Manson, S. S. *Fatigue and Durability of Structural Materials*; ASM International: Materials Park, Ohio, 2006.

REVIEWERS' COMMENTS

Reviewer #1 (Remarks to the Author):

After going through the revised manuscript, the response letter and the PyePAL code available on GitHub, I believe that the authors have satisfactorily addressed all my comments and I am happy to recommend the paper for publication in Nature Communications.

Reviewer #2 (Remarks to the Author):

The authors have addressed all of my concerns, and I now recommend the manuscript for publication. I also would like to note that I appreciated all the work that went into the website for PyePAL especially highlighting the differences between prior e-PAL work and PyePAL (<https://pyepal.readthedocs.io/en/latest/background.html>).